# Chronic Adaptations to Eccentric Cycling Training: A Systematic Review and Meta-Analysis

**DOI:** 10.3390/ijerph20042861

**Published:** 2023-02-06

**Authors:** Renan Vieira Barreto, Leonardo Coelho Rabello de Lima, Fernando Klitzke Borszcz, Ricardo Dantas de Lucas, Benedito Sérgio Denadai

**Affiliations:** 1Human Performance Laboratory, Department of Physical Education, São Paulo State University, Rio Claro 13506-900, Brazil; 2School of Physical Education and Sport of Ribeirão Preto, University of São Paulo, Ribeirão Preto 14040-900, Brazil; 3Physical Effort Laboratory, Sports Centre, Federal University of Santa Catarina, Florianópolis 88040-900, Brazil

**Keywords:** strength, aerobic capacity, COPD, eccentric exercise

## Abstract

This study aimed to investigate the effects of eccentric cycling (ECC_CYC_) training on performance, physiological, and morphological parameters in comparison to concentric cycling (CON_CYC_) training. Searches were conducted using PubMed, Embase, and ScienceDirect. Studies comparing the effect of ECC_CYC_ and CON_CYC_ training regimens on performance, physiological, and/or morphological parameters were included. Bayesian multilevel meta-analysis models were used to estimate the population’s mean difference between chronic responses from ECC_CYC_ and CON_CYC_ training protocols. Group levels and meta-regression were used to evaluate the specific effects of subjects and study characteristics. Fourteen studies were included in this review. The meta-analyses showed that ECC_CYC_ training was more effective in increasing knee extensor strength, vastus lateralis fiber cross-sectional area, and six-minute walking distance compared to CON_CYC_. Moreover, ECC_CYC_ was as effective as CON_CYC_ in decreasing body fat percentage. CON_CYC_ was more effective in increasing $\dot{V}O_{2\max}$ and peak power output attained during concentric incremental tests. However, group-level analyses revealed that ECC_CYC_ was more effective than CON_CYC_ in improving $\dot{V}O_{2\max}$ in patients with cardiopulmonary diseases. ECC_CYC_ is a viable modality for exercise interventions aiming to improve parameters of muscle strength, hypertrophy, functional capacity, aerobic power, and body composition, with more advantages than CON_CYC_ training in improving neuromuscular variables.

## 1. Introduction

Over the last decades, sports scientists and physiologists have advanced our knowledge of the mechanisms and applications of eccentric (i.e., lengthening) muscle contractions. Given its distinctive characteristics, eccentric contractions may induce different morphological, neuromuscular, and metabolic adaptations compared to isometric and concentric muscle actions [1,2]. Evidence from isokinetic exercise indicates that eccentric contractions are more effective in increasing muscle strength than concentric contractions [3]. Additionally, there is robust evidence supporting the utilization of eccentric exercise for the treatment of tendinopathies [4,5,6] and injury prevention and rehabilitation [7,8,9,10,11]. Recently, there has been a growing interest in the utilization of eccentric exercises in the treatment of individuals with poor exercise tolerance, such as elderly individuals and patients with cardiopulmonary diseases [12,13].

In fact, the low-energy cost associated with eccentric muscle work makes eccentric exercises a feasible alternative to counteract sarcopenia and declines in the functional capacity of older individuals, especially those with clinical conditions, since eccentric exercises may provide sufficient stimulus to trigger neuromuscular adaptations without imposing severe cardiovascular and/or respiratory burden [13,14,15,16]. In this context, eccentric exercise via motorized cycle ergometers, named eccentric cycling (ECC_CYC_), comprises a safe way to exercise eccentrically, avoiding impact on lower limb joints and falls, as well as allowing appropriate quantification of negative work absorbed during exercise bouts. ECC_CYC_ can be categorized as a moderate-load eccentric exercise [12] consisting of submaximal eccentric exercise performed continuously for long periods (i.e., 10–30 min). In practice, pedaling eccentrically consists in resisting (i.e., attempting to brake) the motor-driven backward movement of the pedals. Thus, during ECC_CYC_ sessions, one must produce a large number of eccentric contractions with the locomotor muscles, mainly the knee and hip extensor muscles [17], in a coordinated manner, trying to maintain the level of resistive force prescribed, usually displayed on a screen connected to the cycle ergometer.

Interestingly, in addition to the known potent stimulus of eccentric exercises for muscle strength development, chronic ECC_CYC_ interventions can confer improvements in aerobic power, exercise tolerance, and body composition parameters [14,18,19,20,21]. Hence, ECC_CYC_ has the potential to improve distinctive performance and physiological/morphological parameters by inducing both energetic/metabolic and tensional adaptive stimuli within the same session, characterizing a time-effective modality [22]. However, some studies only indicate neuromuscular and morphological muscle adaptations following ECC_CYC_ interventions [23,24,25,26], and there is evidence suggesting that eccentric endurance exercises may not have an impact on aerobic metabolism [27].

These disagreements in the chronic responses to ECC_CYC_ may be related to the lack of information regarding the eccentric exercise intensity continuum, which makes the prescription of ECC_CYC_ load based on a specific level of homeostasis disturbance related to a targeted adaptation difficult. Additionally, the fitness level of the individuals submitted to ECC_CYC_ interventions could explain, in part, the controversial results of ECC_CYC_ adaptations since the moderate intensity-like behavior (for details, please see Barreto et al. [22]) of the physiological responses to ECC_CYC_ may be insufficient to promote aerobic/metabolic adaptations in healthy active and highly trained individuals. Therefore, the present systematic review aimed to determine the effectiveness of ECC_CYC_ interventions in improving different performance, physiological, and morphological parameters, as well as investigating the impact of the intervention and population characteristics on these training outcomes. We choose to compare ECC_CYC_ with concentric cycling (CON_CYC_) interventions since CON_CYC_ protocols are widely used for exercise treatment in clinical populations [28,29] and to improve aerobic and body composition parameters in the general population [30].

## 2. Materials and Methods

The present review is part of a larger systematic review project that aims to investigate the acute and chronic physiological responses to ECC_CYC_ compared to CON_CYC_. The original protocol was prospectively registered within the Open Science Framework (https://osf.io/sa6g3, accessed on 1 December 2022). The search strategy and selection of studies included both transversal and longitudinal studies that investigated the responses to ECC_CYC_ bouts and training protocols, respectively. The present review followed the Preferred Reporting Items for Systematic Reviews and Meta-Analyses (PRISMA) guidelines [31].

### 2.1. Study Eligibility

The PICO framework (Population, Intervention, Comparator, and Outcome) was used to establish the eligibility criteria. The population included male and female human subjects without restrictions on age, health condition, or level of physical fitness. The exercise intervention must have comprised ECC_CYC_ training protocols lasting at least one week. The comparator consisted of CON_CYC_ training of the same duration. Investigations that reported outcomes related to neuromuscular function, aerobic power, functional capacity, and body composition before and after ECC_CYC_ and CON_CYC_ training protocols were considered for inclusion. Only original investigations published in peer-reviewed outlets and written in English were considered eligible. Studies that investigated the responses to single-leg ECC_CYC_ training or ECC_CYC_ performed with the upper limbs were excluded.

### 2.2. Information Sources

The searches were conducted until February 2021 and updated in November 2021 using the online electronic databases PubMed, ScienceDirect, and Embase. Studies meeting the inclusion criteria identified in the reference lists of the included articles were also included in the present review. Missing data and/or information from the selected studies were requested via electronic mailing to the authors.

### 2.3. Search Strategy

The search strategy was designed to track all possible studies on the topic of “eccentric cycling”. Five target studies on this topic were used to formulate the search strategy [24,32,33,34,35]. The title and abstract of the records were examined for the identification of possible search terms using the word frequency analysis tool of the PubMed database.

The search strategy was verified by identifying the recognized relevant studies in the results of preliminary searches and by identifying new relevant studies obtained through changing search terms. Hence, we adopted the following search strategy through titles and abstracts of indexed documents: (“eccentric” OR “eccentrically” OR “negative work”) AND (“cycling” OR “bicycle” OR “pedaling” OR “pedalling” OR “ergometer” OR “ergometry”). No temporal clipping was established.

### 2.4. Study Selection

Two authors (R.V.B. and L.C.R.L.) carried out the selection of studies independently, using a freely available software—Rayyan QCRI (https://www.rayyan.ai/, accessed on 1 December 2022) [36]. Following the exclusion of duplicates, the titles and abstracts were screened, and irrelevant records were removed. Subsequently, the full-text articles were reviewed and assessed for eligibility.

### 2.5. Data Extraction

The data of each study included in this review were extracted separately by two authors (R.V.B. and L.C.R.L) into an Excel spreadsheet (Microsoft, Redmond, WA, USA). The same authors then compared their spreadsheets and addressed the inconsistencies through discussion. When necessary, data were extracted from figures using the freely available software Web Plot Digitizer (https://automeris.io/WebPlotDigitizer, accessed on 1 December 2022).

### 2.6. Data Items

Information about the publication (i.e., author, year, journal, and digital object identifier—DOI), study design (e.g., randomized or quasi-randomized), population (i.e., sample size, age, height, body mass, maximal oxygen uptake ($\dot{V}O_{2\max}$), and health condition), intervention (i.e., intensity, duration, and type of the sessions, number of sessions per week, pedal cadence, and duration of the ECC_CYC_ training protocol), comparator (i.e., intensity, duration, and type of the sessions, number of sessions per week, pedal cadence, and duration of the CON_CYC_ training protocol), and outcomes (i.e., mean and standard deviation (SD) of the performance, physiological, and/or morphological parameter assessed before and after CON_CYC_ and ECC_CYC_ training period and the inferential statistics parameters) was extracted from all studies included in this review.

The terms $\dot{V}O_{2\max}$ and PPO were used to represent measures of maximal and peak oxygen consumption and maximal and peak power output, respectively, obtained from an incremental CON_CYC_ test performed until exhaustion or a symptom-limited incremental CON_CYC_ test.

### 2.7. Study Quality Assessment

The methodological quality of the selected reports was rated using the Physiotherapy Evidence-Based Database (PEDro) scale [37]. The PEDro scale comprises 11 criteria related to the external (item 1) and internal (items 2–9) validity of the study, as well as the statistical procedures (items 10–11). Except for the first item, which was not utilized to generate the PEDro score, the report receives one point for each satisfying item. Therefore, the higher the study score (with 10 being the highest), the higher the study’s quality [37]. Two authors (R.V.B. and L.C.R.L.) assessed the quality of the included studies independently, and discrepancies were handled by a third author (B.S.D.).

### 2.8. Effect Sizes Calculation

Three different effect sizes were calculated for each variable: (1) the mean percent difference between pre- and post-ECC_CYC_ training, (2) the mean percent difference between pre- and post-CON_CYC_ training, and (3) the net effect between training modalities (i.e., the mean difference between pre-to-post ECC_CYC_ and pre-to-post CON_CYC_). Thus, a negative net effect means that the change induced by ECC_CYC_ training in the analyzed variable was % greater than the change induced by CON_CYC_ training and vice versa.

The precision of each effect size was determined by the standard error (SE) of the pre-to-post change for CON_CYC_ and ECC_CYC_ conditions. The SE of the change for each condition (i.e., CON_CYC_ and ECC_CYC_) was calculated by dividing the SD of the pre-to-post change by the square root of the sample size. The SE of the net difference between CON_CYC_ and ECC_CYC_ was calculated as the sum of the pre-to-post SE of each condition via their variances as follows:$${\mathrm{SE}}_{\mathrm{net}\;\mathrm{difference}}=\sqrt{\left({\left({\mathrm{SE}}_{{\mathrm{CON}}_{\mathrm{CYC}}}\right)}^{2}+{\left({\mathrm{SE}}_{{\mathrm{ECC}}_{\mathrm{CYC}}}\right)}^{2}\right)}$$

Note that SE of net difference was derived to present to the reader the “observed” effect sizes in the studies; the net mean difference between the conditions was determined during the statistical analysis, as detailed below. Within the meta-analytic model, the weight of each effect size was set by the inverse of the squared SE (1/SE^2^). Thus, effects derived from studies with less variability in responses and/or a greater number of participants exerted greater weight on meta-analyzed effect size. The SD of the pre-to-post change was determined using the exact *p*-values via t-statistics, confidence intervals, F-values [38], and raw data, or was determined from data extracted from figures. When it was not possible, the SD of the change was inputted by the mean correlation coefficient (r) derived from pre-to-post scores of the studies in which the inferential statistics deemed its determination possible [38]. A moderate correlation coefficient (r = 0.50) was adopted for the imputation of SD of the change when none of the alternatives described above was possible.

Pre-to-post effect sizes and their respective SE were converted to percentage units by dividing by the mean of CON_CYC_ and ECC_CYC_ conditions, respectively, and multiplying it by 100.

### 2.9. Statistical Analysis

The meta-analyses were conducted within a Bayesian framework using multilevel models. Analyses were performed in the statistical software R (v4.0; R Core Team [2020], Vienna, Austria) in its graphical interface RStudio (v1.2.5; Boston, MA, USA). The brms package [39] was used for analyses, which allowed the adjustment of multilevel Bayesian models using Stan (i.e., a platform for statistical modeling and high-performance statistical computation) [40].

We derived the effects for each arm to verify the pre-to-post change for both CON_CYC_ and ECC_CYC_ interventions and evaluate the contrast effect between the two conditions. Hence, two analysis models were carried out for each outcome. The first model was composed of a linear meta-regression analysis, where the response variable was the effect sizes and the covariate (i.e., model fixed effect) was the ECC_CYC_ and CON_CYC_ conditions coded numerically as 0 and 1, respectively. *Ergo*, the intercept of the meta-regression provided the population (fixed effect) meta-analyzed average effect of the ECC_CYC_ condition, and the slope of the regression provided the average net difference between ECC_CYC_ and CON_CYC_ conditions. The second model included only the pre-to-post effect sizes of CON_CYC_ to derivate the population’s meta-analyzed average effect of this condition. Random effects in the first model included the intercept and slope of each study for each time point in the case of repeated measures (i.e., each pair of CON_CYC_ and ECC_CYC_ effect sizes for each time point), the intercept and slope of each health condition of the participants, and the duration of the interventions (i.e., specific group-levels). For the second model, random effects were set for each effect size, the health condition of the participants, and the duration of the intervention identities (intercepts).

When the outcome provided more than ten included studies, a meta-regression with more than the two conditions as covariates was carried out [38]. Fixed effects (i.e., covariates) included the conditions (ECC_CYC_ and CON_CYC_), the duration of the intervention (linear as days), the interaction conditions × duration of the intervention, and participants’ $\dot{V}O_{2\max}$ (linear as mL/kg/min). Random effects included the intercept, slope of condition, slope of intervention duration, and slope interaction conditions × duration of the intervention.

Weakly-informative Student’s *t* prior distributions (df = 3, µ = 0, and σ = 10) were used for fixed-effects models, and half Student’s *t*-distributions (df = 3 and σ = 10) were used for between-group-level variance effects (i.e., τ values). Model fitting was performed using Markov Chain Monte Carlo (MCMC) methods, more specifically, the No-U-Turn sampler (NUTS) implemented in Stan. For each model, four chains were run in parallel with 4000 iterations and a warm-up of 1000 iterations. The convergence of the models was verified with Gelman–Rubin diagnostics (R) [41].

To deal with repeated measures in the meta-analyses of studies with more than one effect for the same participant, variance-covariance matrices were calculated. When the information provided in the studies was insufficient to determine the correlation between the dependent effect sizes to perform the matrix calculation, a moderate coefficient of correlation (r = 0.50) was assumed between the effects derived from the same participants. Furthermore, sensitivity analyzes were also performed using the values of r = 0.30 and 0.70 in the calculation of matrices (see Appendix A, which presents the results of sensitivity analyses). For meta-regressions in which the covariate $\dot{V}O_{2\max}$ was missing, we input the data during model fitting using a multivariate model as described elsewhere [42].

Heterogeneity between the effects and group levels was presented as SD (tau-τ). All posterior data generated by MCMC were reported as medians with two-tailed 95% credible intervals (CrI). Furthermore, considering the complete posterior distributions, the probability (in %) of the effect being greater than zero (*p* > 0) was presented; that is, the area of the posterior distribution located above zero. The area of the posterior distribution of pre-to-post training effect sizes located above zero indicates the probability of the training modality (i.e., CON_CYC_ or ECC_CYC_), inducing an increase in the variable from pre-training measures. The area of the posterior distribution of net effects between training modalities located above zero indicates the probability of the CON_CYC_ inducing a greater increase in the variable and vice versa.

## 3. Results

### 3.1. Study Selection

The literature search yielded a total of 992 results. A total of 628 titles and abstracts were screened after duplicate removal, and 105 full-text articles were read and assessed for eligibility. Four additional articles were retrieved from reference lists. A total of 14 articles were included in this review (Figure 1).

### 3.2. Study Characteristics

Table 1 summarizes the main characteristics of the included studies. Most of the studies (57%) included in this systematic review recruited patients with cardiopulmonary diseases. Specifically, three studies recruited coronary artery disease patients [19,34,43], two studies enrolled chronic heart failure patients [18,44], and three studies reported data from chronic obstructive pulmonary disease patients [14,35,45]. Among the remaining studies, two have investigated the effects of ECC_CYC_ and CON_CYC_ training in healthy individuals [24,46], two recruited sedentary participants [23,25], one evaluated amateur cyclists [26], and one evaluated obese adolescents [20].

The training period adopted in the included studies ranged from 5 to 12 weeks (mode = 8 weeks), with weekly frequencies varying between 2 and 5 sessions per week (mode = 3 sessions per week). The duration of training sessions ranged between 10 and 30 min (mode = 30 min). Except for two studies [26,45], which used interval training sessions, all others used fixed-intensity sessions performed continuously (i.e., without rest intervals).

The intensity of ECC_CYC_ and CON_CYC_ training sessions of the studies included in this review was prescribed based on individuals’ rate of perceived exertion (RPE), heart rate (HR) corresponding to the ventilatory threshold, percentage of PPO, $\dot{V}O_{2\max}$, or HR_max_ attained during the CON_CYC_ incremental test and percentage of age-predicted HR_max_. In 14% of the included studies, the intensity of the sessions was defined so that the participants of the two groups (i.e., ECC_CYC_ and CON_CYC_) exercised at the same PO [23,25]; in 22% of the studies, the training intensity was chosen to match VO_2_ during the ECC_CYC_ and CON_CYC_ sessions [20,34,46]; and 36% of the studies set the intensity of both training groups to elicit the same relative HR [14,19,24,35,43]. Only two studies (14%) matched ECC_CYC_ and CON_CYC_ training intensity with the same RPE [26,45], and two studies (14%) prescribed training intensities based on safe RPE and HR values for the investigated population without establishing equalization criteria between groups [18,44].

### 3.3. Quality Assessment

The scores on the PEDro scale ranged from 3 to 8 points (mode = 6 points; mean = 6 points) (Table 2). Most studies (79%) included in this review presented “good” (i.e., 6–8 points) methodological quality, while two studies (14%) presented “fair” (i.e., 4–5 points) methodological quality and one study (7%) was rated with “poor” methodological quality [47]. Most studies (86%) included in this review were randomized controlled clinical trials, except for two studies [25,45], which adopted a quasi-randomized design, where participants were allocated into groups according to their forced expiratory air volume in the first second (FEV_1_) and age [45], and maximal isometric voluntary force of the knee extensor muscles [25]. None of the studies scored on criteria 5 and 6 of the scale (i.e., criteria related to blinding of subjects and therapists who administered the intervention, respectively). Blinding of assessors was performed in five studies, which scored at criterion 7 [14,19,20,35,44].

### 3.4. Meta-Analyses

#### 3.4.1. Isometric Peak Torque (IPT)

Eleven of the included studies assessed knee extensors IPT (n participants = 212) (Figure 2). The estimated average effect of the difference between pre- and post-CON_CYC_ training IPT values showed that CON_CYC_ was effective in improving IPT (µ = 5.83% [3.01%, 8.79%]; *p* > 0 = 100%) (Figure 2A). The estimated average effect of the difference between pre- and post-ECC_CYC_ training also indicated that ECC_CYC_ was effective in improving IPT (µ = 13.02% [9.19%, 17.17%]; *p* > 0 = 100%) (Figure 2B). The meta-analyzed net effect showed a more favorable effect of ECC_CYC_ than CON_CYC_ on IPT (µ = −6.82% [−11.16%, −2.82%]) (Figure 2C). The posterior density of the average net effect indicated no probability (*p* > 0 = 0%) of the CON_CYC_ inducing greater improvements in IPT compared to ECC_CYC_ training. The heterogeneity between CON_CYC_ training effects was lower (τ = 4.58% [3.15%, 6.78%]; Figure 2D) than the heterogeneity between ECC_CYC_ training effects (τ = 7.55% [5.47%, 10.8%]; Figure 2E) and heterogeneity between net effects (τ = 5.9% [3.52%, 9.21%]; Figure 2F).

Figure 3 presents the conditional effects, the modifying effects, and the residual heterogeneity derived from the meta-regression. The effects on knee extensors IPT were modified mainly by the duration of the intervention. The meta-regression revealed that the difference between ECC_CYC_ and CON_CYC_ training effects on IPT becomes greater as the intervention duration increases. The aerobic fitness expressed as the relative $\dot{V}O_{2\max}$ (i.e., mL/kg/min) showed no important modification to the changes in IPT following the two types of cycling training. Small to moderate heterogeneity was observed in the meta-regression-derived effects.

#### 3.4.2. Isokinetic Concentric Peak Torque (ICPT)

Four studies investigated the effects of ECC_CYC_ and CON_CYC_ training on knee extensors ICPT (n participants = 82) (see Appendix A). The meta-analyzed average effect of pre-to-post CON_CYC_ training effects indicated that CON_CYC_ did not affect ICPT (µ = −0.92% [−9.42%, 6.51%]; *p* > 0 = 39.63%). On the other hand, the estimated average effect of pre-to-post ECC_CYC_ training effects showed that ECC_CYC_ was effective in improving ICPT (µ = 8.82% [−11.49%, 29.72%]; *p* > 0 = 83.33%). The meta-analyzed net effect showed a more favorable effect of ECC_CYC_ than CON_CYC_ training on ICPT (µ = −9.85% [−38.42%, 15.93%]). The posterior density of the average net effect indicated a low probability (*p* > 0 = 18.2%) of the CON_CYC_, inducing greater improvements in ICPT compared to ECC_CYC_ training. The heterogeneity between pre-to-post CON_CYC_ training effects was lower (τ = 3.62% [0.2%, 12.4%]) compared to heterogeneity between pre-to-post ECC_CYC_ training effects (τ = 7.56% [0.35%, 27.8%]) and heterogeneity between net effects (τ = 6.79% [0.25%, 27.2%]).

Group-level effects of intervention duration indicated that CON_CYC_ training protocols lasting 1 to 2 months impaired ICPT (µ = −2.13% [−13.6%, 6.1%]; *p* > 0 = 29.98%), while protocols lasting longer than 2 months did not affect ICPT (µ = −0.004% [−7.2%, 7.51%]; *p* > 0 = 49.9%). Group-level analysis of intervention duration indicated similar positive effects between ECC_CYC_ training protocols lasting 1 to 2 months (µ = 8.8% [−12.5%, 31.3%]; *p* > 0 = 82.75%) and protocols lasting longer than 2 months (µ = 8.21% [−10.8%, 27.0%]; *p* > 0 = 85.23%). Considering the effect of intervention duration on the net effect between training modalities, the superiority in improving the ICPT of ECC_CYC_ compared to CON_CYC_ was pronounced in the studies with training protocols lasting 1 to 2 months (µ = −13.0% [−36.6%, 10.4%]; *p* > 0 = 10.53%) compared to those using protocols lasting longer than 2 months (µ = −6.74% [−27.2%, 11.8%]; *p* > 0 = 17.4%). There was low heterogeneity between group-level pre-to-post CON_CYC_ effects of intervention duration (τ = 3.18% [0.16%, 14.2%]) and considerable heterogeneity between group-level pre-to-post ECC_CYC_ effects (τ = 6.52% [0.35%, 33.2%]) and between group-level net effects (τ = 8.81% [0.33%, 38.4%]).

Group-level effects of the population showed that CON_CYC_ training was similarly ineffective in improving the ICPT of obese adolescents (µ = −0.42% [−10.5%, 9.56%]; *p* > 0 = 45.63%) and patients with cardiopulmonary diseases (µ = −0.16% [−8.72%, 8.37%]; *p* > 0 = 47.6%), and may have been prejudicial to amateur cyclists (µ = −2.43% [−12.4%, 5.86%]; *p* > 0 = 27.08%). Moreover, the group-level analysis indicated similar positive effects of ECC_CYC_ training in obese adolescents (µ = 12.7% [−9.23%, 34.0%]; *p* > 0 = 90.0%) and patients with cardiopulmonary diseases (µ = 13.0% [−7.15%, 32.8%]; *p* > 0 = 92.98%), while the estimated group-level effect for amateur cyclists indicated that ECC_CYC_ training was ineffective in increasing ICPT (µ = −0.58% [−20.4%, 25.5%]; *p* > 0 = 47.85%). Group-level net effects of the population indicated that ECC_CYC_ was more effective in improving ICPT than CON_CYC_ training for all sub-groups, but the differences between training modalities were more pronounced in obese adolescents (µ = −12.3% [−40.7%, 13.7%]; *p* > 0 = 12.58%) and patients with cardiopulmonary diseases (µ = −12.4% [−38.2%, 13.2%]; *p* > 0 = 11.48%) compared to amateur cyclists (µ = −4.79% [−33.5%, 21.6%]; *p* > 0 = 32.1%). Low heterogeneity was found between group-level pre-to-post CON_CYC_ effects of the population (τ = 3% [0.12%, 11.9%]), while considerable heterogeneity was found between group-level pre-to-post ECC_CYC_ effects (τ = 10.6% [0.74%, 34.6%]) and between group-level net effects (τ = 7.53% [0.4%, 30%]).

#### 3.4.3. Isokinetic Eccentric Peak Torque (IEPT)

Three of the included studies investigated the effects of ECC_CYC_ and CON_CYC_ training on knee extensors IEPT (n participants = 58) (see Appendix A). The estimated average effect of the difference between pre- and post-CON_CYC_ training IEPT values showed that CON_CYC_ increased IEPT (µ = 2.26% [−1.55%, 5.83%]; *p* > 0 = 91.28%). The meta-analyzed average effect of the difference between pre- and post-ECC_CYC_ training IEPT values also showed a positive effect of ECC_CYC_ on IEPT (µ = 9.91% [4.47%, 14.66%]; *p* > 0 = 99.9%). The estimated net effect indicated that ECC_CYC_ was more effective in increasing IEPT than CON_CYC_ (µ = −9.17% [−16.33%, −2.16%]). Moreover, the posterior distribution of the average net effect showed a very low probability (*p* > 0 = 1.1%) of CON_CYC_ inducing greater improvements in IEPT than ECC_CYC_ training. The heterogeneity was low and similar between pre-to-post CON_CYC_ training effects (τ = 1.19% [0.05%, 5.02%]), pre-to-post ECC_CYC_ training effects (τ = 1.44% [0.06%, 5.83%]), and net effects (τ = 1.45% [0.08%, 6.75%]).

Group-level effects of intervention duration showed similar effects between CON_CYC_ training protocols lasting 1 to 2 months (µ = 2.26% [−2.34%, 6.65%]; *p* > 0 = 86.93%) and protocols lasting longer than 2 months (µ = 2.25% [−1.27%, 5.45%]; *p* > 0 = 90.98%) on IEPT. Group-level analysis of intervention duration also indicated similar effects between ECC_CYC_ training protocols lasting 1 to 2 months (µ = 9.64% [2.89%, 14.8%]; *p* > 0 = 99.55%) and protocols lasting longer than 2 months (µ = 11.1% [6.62%, 15.5%]; *p* > 0 = 100%). Similarly, group-level analysis of intervention duration indicated equivalent net effects between investigations using training protocols lasting 1 to 2 months (µ = −8.73% [−15.0%, −0.91%]; *p* > 0 = 1.68%) and protocols lasting longer than 2 months (µ = −9.26% [−14.9%, −3.81%]; *p* > 0 = 0.23%). The heterogeneity was low and similar between pre-to-post CON_CYC_ training effects (τ = 1.32% [0.06%, 5.8%]), pre-to-post ECC_CYC_ training effects (τ = 1.83% [0.09%, 7.24%]), and net effects (τ = 1.72% [0.08%, 7.59%]).

#### 3.4.4. Vastus Lateralis Fiber Cross-Sectional Area (f-CSA)

Three of the included studies investigated the effects of ECC_CYC_ and CON_CYC_ training on f-CSA (n participants = 40) (see Appendix A). The meta-analyzed average effect of the difference between pre- and post-CON_CYC_ training f-CSA values indicated that CON_CYC_ was effective in increasing f-CSA (µ = 14.54% [4.56%, 24.43%]; *p* > 0 = 99.48%). The estimated average effect of the difference between pre- and post-ECC_CYC_ training also indicated that ECC_CYC_ was effective in increasing f-CSA (µ = 13.87% [−9.5%, 38.18%]; *p* > 0 = 90.33%). The meta-analyzed net effect showed a more favorable effect of ECC_CYC_ than CON_CYC_ on f-CSA (µ = −2.55% [−38.51%, 32.68%]). The posterior density of the average net effect showed a slightly smaller probability (*p* > 0 = 43.7%) of CON_CYC_ inducing greater increases in f-CSA compared to ECC_CYC_ training. The heterogeneity between CON_CYC_ training effects was lower (τ = 3.48% [0.18%, 14.0%]) compared to the heterogeneity between ECC_CYC_ training effects (τ = 15.3% [2.58%, 38.1%]) and heterogeneity between net effects (τ = 16.7% [1.76%, 43.8%]), which were considerably high.

The estimated group-level effects of intervention duration indicated similar effects of CON_CYC_ training protocols lasting 1 to 2 months (µ = 14.0% [2.99%, 24.8%]; *p* > 0 = 99.05%) and protocols lasting longer than 2 months (µ = 14.5% [1.46%, 28.1%]; *p* > 0 = 98.3%) on f-CSA. Moreover, group-level analyses showed greater increases in f-CSA following ECC_CYC_ training protocols lasting 1 to 2 months (µ = 16.3% [−5.8%, 39.3%]; *p* > 0 = 93.78%) than following protocols lasting longer than 2 months (µ = 12.3% [−12.7%, 37.8%]; *p* > 0 = 86.53%). Group-level analyses of net effects indicated that investigations using training protocols lasting 1 to 2 months presented more favorable effects following ECC_CYC_ than CON_CYC_ (µ = −3.83% [−34.4%, 26.0%]; *p* > 0 = 38.43%), while those using protocols lasting longer than 2 months presented no difference between ECC_CYC_ and CON_CYC_ (µ = −0.43% [−35.8%, 34.6%]; *p* > 0 = 49.05%). The heterogeneity between group-level pre-to-post CON_CYC_ effects was lower (τ = 3.54% [0.14%, 15.5%]) than the heterogeneity between ECC_CYC_ training effects (τ = 7.13% [0.26%, 30.7%]) and the heterogeneity between net effects (τ = 8.13% [0.41%, 35.6%]).

#### 3.4.5. Peak Power Output (PPO)

Ten of the included studies investigated the effects of ECC_CYC_ and CON_CYC_ training on PPO (n participants = 215) (Figure 4). The estimated average effect of the difference between pre- and post-CON_CYC_ training PPO values indicated that CON_CYC_ was effective in increasing PPO (µ = 16.8% [11.43%, 23.13%]; *p* > 0 = 100%) (Figure 4A). The average effect of the difference between pre- and post-ECC_CYC_ training also indicated that ECC_CYC_ was effective in increasing PPO (µ = 10.51% [5.36%, 16.12%]; *p* > 0 = 100%) (Figure 4B). The meta-analyzed net effect showed a more favorable effect of CON_CYC_ than ECC_CYC_ training on PPO (µ = 6.88% [2.34%, 11.02%]) (Figure 4C). The posterior distribution of the average net effect showed a greater probability (*p* > 0 = 99.55%) of the CON_CYC_ inducing greater increases in PPO compared to ECC_CYC_ training. There was considerable heterogeneity between CON_CYC_ training effects (τ = 8.03% [4.92%, 14.0%]; Figure 4D) and between ECC_CYC_ training effects (τ = 7.41% [4.71%, 13.0%]; Figure 4E) and moderate heterogeneity between net effects (τ = 4.44% [0.88%, 10.1%]; Figure 4F).

Figure 5 presents the conditional effects, the modifying effects, and the residual heterogeneity derived from the meta-regression. The effects on PPO were modified mainly by the duration of the intervention and the subjects’ aerobic fitness expressed as the relative $\dot{V}O_{2\max}$. The meta-regression revealed that the difference between ECC_CYC_ and CON_CYC_ effects on PPO becomes smaller as the intervention duration increases. Moreover, PPO changes following both training modalities were greater in subjects with lower relative $\dot{V}O_{2\max}$ values. Small to moderate heterogeneity was observed in the meta-regression-derived effects.

#### 3.4.6. Maximal Oxygen Uptake ($\dot{V}O_{2\max}$)

Nine of the included studies investigated the effects of ECC_CYC_ and CON_CYC_ training on $\dot{V}O_{2\max}$ (n participants = 189) (Figure 6). The meta-analyzed average effect of the difference between pre- and post-CON_CYC_ training $\dot{V}O_{2\max}$ values indicated that CON_CYC_ was effective in increasing $\dot{V}O_{2\max}$ (µ = 6.51% [0.52%, 12.34%]; *p* > 0 = 98.13%) (Figure 6A). Similarly, the estimated average effect of the difference between pre- and post-ECC_CYC_ training $\dot{V}O_{2\max}$ values indicated that ECC_CYC_ was effective in increasing $\dot{V}O_{2\max}$ (µ = 4.06% [−7.44%, 14.68%]; *p* > 0 = 79.5%) (Figure 6B). The meta-analyzed net effect showed that CON_CYC_ was more effective in increasing $\dot{V}O_{2\max}$ than ECC_CYC_ (µ = 1.64% [−11.6%, 13.51%]) (Figure 6C). The posterior distribution of the average net effect indicated a greater probability (*p* > 0 = 62.53%) of the CON_CYC_ inducing greater improvements in $\dot{V}O_{2\max}$ compared to ECC_CYC_ training. The heterogeneity was similar between pre-to-post CON_CYC_ training effects (τ = 3.27% [0.23%, 8.61%]; Figure 6D), pre-to-post ECC_CYC_ training effects (τ = 2.82% [0.13%, 10.8%]; Figure 6E), and net effects (τ = 2.99% [0.16%, 9.65%]; Figure 6F).

Group-level effects of intervention duration showed similar effects of CON_CYC_ training protocols lasting 1 to 2 months (µ = 6.86% [1.15%, 11.4%]; *p* > 0 = 98.8%) and protocols lasting longer than 2 months (µ = 5.91% [−1.06%, 11.4%]; *p* > 0 = 95.75%) on $\dot{V}O_{2\max}$ (Figure 6G). Group-level analysis showed similar effects of ECC_CYC_ training protocols lasting 1 to 2 months (µ = 4.43% [−4.69%, 14.8%]; *p* > 0 = 84.23%) and protocols lasting longer than 2 months (µ = 3.30% [−7.49%, 13.0%]; *p* > 0 = 74.93%) (Figure 6H). Interestingly, group-level analysis of net effects showed that investigations using training protocols lasting 1 to 2 months presented more favorable effects following CON_CYC_ than ECC_CYC_ (µ = 2.98% [−5.91%, 11.2%]; *p* > 0 = 77%), but those using protocols lasting longer than 2 months presented no difference between cycling modalities (µ = −0.06% [−9.97%, 9.22%]; *p* > 0 = 49.5%) (Figure 6I). Small to moderate heterogeneity was found between pre-to-post CON_CYC_ training effects (τ = 2.27% [0.09%, 10.40%]; Figure 6J), pre-to-post ECC_CYC_ training effects (τ = 3.56% [0.15%, 18.0%]; Figure 6K), and net effects (τ = 4.46% [0.26%, 20.5%]; Figure 6L).

Group-level effects of the population showed that CON_CYC_ training was effective in improving $\dot{V}O_{2\max}$ in sedentary individuals (µ = 8.77% [1.71%, 16.1%]; *p* > 0 = 99.1%), obese adolescents (µ = 8.39% [1.2%, 17.9%]; *p* > 0 = 98.8%), healthy individuals (µ = 2.36% [−12.6%, 10.7%]; *p* > 0 = 63.83%), patients with cardiopulmonary diseases (µ = 6.82% [0.29%, 12.9%]; *p* > 0 = 97.83%), and amateur cyclists (µ = 5.73% [−1.28%, 13.5%]; *p* > 0 = 95.3%), with sedentary and obese adolescents presenting the greatest improvements (Figure 6M). Group-level analysis indicated that ECC_CYC_ training was effective in improving $\dot{V}O_{2\max}$ in obese adolescents (µ = 13.0% [−0.44%, 25.6%]; *p* > 0 = 97.2%), patients with cardiopulmonary diseases (µ = 8.26% [−2.39%, 17.0%]; *p* > 0 = 94.98%), and amateur cyclists (µ = 4.67% [−4.9%, 15.5%]; *p* > 0 = 86.6%), but was ineffective in sedentary (µ = −0.71% [−12.2%, 10.2%]; *p* > 0 = 42.75%) and may have prejudicial effects in healthy individuals (µ = −8.23% [−21.9%, 8.27%]; *p* > 0 = 16.68%) (Figure 6N). Group-level net effects on the population indicated that CON_CYC_ was more effective in improving $\dot{V}O_{2\max}$ than ECC_CYC_ training in sedentary participants (µ = 8.17% [−6.27%, 19.8%]; *p* > 0 = 89.58%), healthy individuals (µ = 1.68% [−14.4%, 15.8%]; *p* > 0 = 60.05%), and amateur cyclists (µ = 1.45% [−10.2%, 14.0%]; *p* > 0 = 62.45%). However, group-level net effects of the population indicated that ECC_CYC_ was more effective in improving $\dot{V}O_{2\max}$ than CON_CYC_ training in obese adolescents (µ = −1.10% [−14.9%, 12.2%]; *p* > 0 = 42.48%) and patients with cardiopulmonary diseases (µ = −2.17% [−15.6%, 8.85%]; *p* > 0 = 30.8%) (Figure 6O). There was moderate heterogeneity between pre-to-post CON_CYC_ training effects (τ = 3.81% [0.18%, 11.5%]; Figure 6P) and considerable heterogeneity between pre-to-post ECC_CYC_ training effects (τ = 9.12% [1.16%, 20.9%]; Figure 6Q) and net effects (τ = 5.98% [1.03%, 15.3%]; Figure 6R).

#### 3.4.7. Six-Minute Walking Distance (6MWD)

Five of the included studies investigated the effects of ECC_CYC_ and CON_CYC_ training on 6MWD (n participants = 130) (Figure 7). The average effect of the pre-to-post training effects showed that both cycling modalities were effective in increasing 6MWD (CON_CYC_: µ = 8.21% [4.77%, 11.59%]; *p* > 0 = 99.95%, and ECC_CYC_: µ = 10.59% [6.88%, 14.72%]; *p* > 0 = 100%) (Figure 7A,B). The estimate net effect showed that ECC_CYC_ was more effective in increasing 6MWD than CON_CYC_ (µ = −1.98% [−7.82%, 4.09%]) (Figure 7C). The posterior density of the average net effect indicated a low probability (*p* > 0 = 22.5%) of the CON_CYC_ inducing greater increases in 6MWD than ECC_CYC_ training. The heterogeneity was similar between pre-to-post CON_CYC_ training effects (τ = 1.06% [0.05%, 4.22%]; Figure 7D), pre-to-post ECC_CYC_ training effects (τ = 1.18% [0.06%, 4.72%]; Figure 7E), and net effects (τ = 1.27% [0.06%, 5.49%]; Figure 7F).

Group-level effects of intervention duration showed similar effects of CON_CYC_ training protocols lasting 1 to 2 months (µ = 8.23% [5.48%, 10.9%]; *p* > 0 = 100%) and protocols lasting longer than 2 months (µ = 8.35% [4.11%, 13.0%]; *p* > 0 = 99.93%) on 6MWD (Figure 7G). Group-level analysis showed similar effects of ECC_CYC_ training protocols lasting 1 to 2 months (µ = 10.6% [7.57%, 14.0%]; *p* > 0 = 100%) and protocols lasting longer than 2 months (µ = 10.4% [7.10%, 14.2%]; *p* > 0 = 100%) (Figure 7H). Group-level analysis of net effects showed that investigations using training protocols lasting 1 to 2 months presented a pronounced difference (µ = −2.14% [−6.40%, 1.96%]; *p* > 0 = 14.5%) between CON_CYC_ and ECC_CYC_ training responses compared to investigations using protocols lasting longer than 2 months (µ = −1.86% [−7.78%, 4.59%]; *p* > 0 = 26.45%) (Figure 7I). Small heterogeneity was observed between pre-to-post CON_CYC_ training effects (τ = 1.31% [0.05%, 5.93%]; Figure 7J), pre-to-post ECC_CYC_ training effects (τ = 1.39% [0.07%, 6.37%]; Figure 7K), and net effects (τ = 1.78% [0.09%, 8.15%]; Figure 7L).

#### 3.4.8. Body Fat Percentage (BF%)

Three of the included studies investigated the effects of ECC_CYC_ and CON_CYC_ training on BF% (n participants = 50) (see Appendix A). The estimated average effect of the pre-to-post training effects showed that CON_CYC_ was effective in decreasing BF% (µ = −1.11% [−3.91%, 1.52%]; *p* > 0 = 13.58%), and also ECC_CYC_ training (µ = −1.29% [−5.38%, 2.22%]; *p* > 0 = 20.18%). The meta-analyzed net effect indicated that CON_CYC_ and ECC_CYC_ training were similarly effective in decreasing BF% (µ = −0.11% [−4.14%, 4.88%]; *p* > 0 = 47.28%). The heterogeneity between pre-to-post ECC_CYC_ training effects (τ = 2.23% [0.17%, 6.67%]) was slightly greater than the heterogeneity between pre-to-post CON_CYC_ training effects (τ = 1.14% [0.06%, 4.49%]) and net effects (τ = 1.26% [0.05%, 5.14%]).

Group-level effects of intervention duration showed similar effects of CON_CYC_ training protocols lasting 1 to 2 months (µ = −1.10% [−4.02%, 1.33%]; *p* > 0 = 11.65%) and protocols lasting longer than 2 months (µ = −1.13% [−3.92%, 0.81%]; *p* > 0 = 7.25%) on BF%. Group-level analysis indicated that ECC_CYC_ training protocols lasting longer than 2 months (µ = −1.91% [−6.0%, 1.52%]; *p* > 0 = 9.48%) were more effective in decreasing BF% than protocols lasting 1 to 2 months (µ = −0.59% [−5.41%, 3.2%]; *p* > 0 = 35.78%). Group-level analysis of net effects showed that investigations using training protocols lasting 1 to 2 months presented more favorable effects following CON_CYC_ than ECC_CYC_ (µ = −0.85% [−3.73%, 3.19%]; *p* > 0 = 26.65%), but those using protocols lasting longer than 2 months presented more favorable effects following ECC_CYC_ than CON_CYC_ (µ = 0.44% [−2.51%, 3.60%]; *p* > 0 = 65.7%). Small heterogeneity was observed between pre-to-post CON_CYC_ training effects (τ = 0.91% [0.05%, 4.96%]), pre-to-post ECC_CYC_ training effects (τ = 1.50% [0.09%, 6.14%]), and net effects (τ = 1.50% [0.10%, 6.18%]).

## 4. Discussion

The main aim of the present systematic review was to provide accurate estimates of the differences between ECC_CYC_ and CON_CYC_ training effects on performance, physiological, and morphological parameters. The meta-analytic results showed that ECC_CYC_ training was more effective in increasing IPT (µ = −7% [95% CrI −11%, −3%]), ICPT (µ = −10% [95% CrI −38%, 16%]), IEPT (µ = −9% [95% CrI −16%, −2%]), f-CSA (µ = −3% [95% CrI −39%, 33%]), and 6MWD (µ = −2% [95% CrI −8%, 4%]), similarly effective in decreasing BF% (µ = 0% [95% CrI −4%, 5%]), and less effective in increasing PPO (µ = 7% [95% CrI 2%, 11%]) and $\dot{V}O_{2\max}$ (µ = 2% [95% CrI −12%, 14%]) compared to CON_CYC_ training. Despite similar or lesser effectiveness of ECC_CYC_ compared to CON_CYC_ in improving a few variables analyzed in this review, the posterior densities of ECC_CYC_ training effects revealed high probabilities (80–100%) of this modality affecting all the investigated variables positively. Hence, ECC_CYC_ training can be considered a feasible exercise modality for improving parameters related to muscle strength and hypertrophy, aerobic power, exercise tolerance, and body fat content, with advantages in improving strength variables compared to CON_CYC_.

It is well established that the high levels of force produced during eccentric contractions constitute an optimal stimulus for muscle strength development [3,48]. Moreover, evidence suggests that strength gains induced by eccentric exercises are very specific to the type of contraction as well as the movement velocity produced during training sessions [3]. In the meta-analysis published by Roig et al. [3], they compared the effects of eccentric and concentric training on muscle strength, and it was found that eccentric training was more effective in improving total strength (i.e., an average of isometric, concentric, and eccentric maximal force) and eccentric strength, but not concentric and isometric strength. Our results corroborate with previous literature reporting strength adaptations following eccentric and concentric training and showing greater effectiveness of ECC_CYC_ compared to CON_CYC_ in increasing muscle strength. It is important to emphasize that, in most of the studies included in the meta-analyses of strength variables, the workload used during ECC_CYC_ was greater than that used during CON_CYC_ training sessions, which may explain the superior strength gains observed for ECC_CYC_ compared to CON_CYC_ [1,3]. Furthermore, we found greater effectiveness of ECC_CYC_ in improving muscle strength from all modes of contraction (i.e., isometric, concentric, and eccentric), which contradicts previous results suggesting poor transferability of strength gains induced by eccentric exercises to different types of contraction [3,49,50]. Nevertheless, the posterior distribution of average net effects on IPT, ICPT, and IEPT indicated probabilities of 82% to 100% of ECC_CYC_ inducing greater improvements in muscle strength than CON_CYC_ training. Moreover, there was evidence that the superiority of ECC_CYC_ in increasing IPT compared to CON_CYC_ training becomes greater as intervention duration increases. Therefore, the existing data indicate that the prescription of ECC_CYC_ aiming to develop locomotor muscle strength is more advantageous in comparison to CON_CYC_ training.

Previous systematic reviews indicated that eccentric exercise might generate greater muscle hypertrophy than concentric exercise due to the greater mechanical tension imposed on muscle fibers during eccentric actions [3,51]. Accordingly, the present results showed that ECC_CYC_ was more effective than CON_CYC_ training in increasing f-CSA. It is important to note that the estimated average effect showed a small difference (µ = −3%) between changes induced by ECC_CYC_ and CON_CYC_ and similar probabilities for each modality to induce greater changes than the other (i.e., 44% of probability of the CON_CYC_ being more effective and 56% of probability of the ECC_CYC_ being more effective). A possible explanation for the discrepancy observed between the results of the present study (which indicate a small difference between muscle hypertrophy induced by ECC_CYC_ vs. CON_CYC_ training), and the current literature (which suggests the superiority of eccentric contractions in producing muscle hypertrophy) could be the distinctive characteristics of muscle remodeling to eccentric and concentric muscle overloading [52]. It has been shown that changes in muscle size are associated with changes in f-CSA following concentric exercises, but muscle growth may occur without significant changes in f-CSA following eccentric exercises. Instead, muscle hypertrophy following eccentric exercises may occur through increased sarcomere length and/or the addition of sarcomeres in series [21,52,53,54]. Unfortunately, the only measurement of muscle hypertrophy that was possible to meta-analyze was f-CSA due to the lack of use of more reliable methods to evaluate muscle size (e.g., magnetic resonance image, computerized tomography, or ultrasonography) in investigations assessing muscle hypertrophy following ECC_CYC_ and CON_CYC_. Hence, more studies are needed to establish the effectiveness of ECC_CYC_ in inducing muscle hypertrophy in comparison to CON_CYC_.

Unlike most eccentric exercise modalities, ECC_CYC_ is performed at submaximal intensities, with large muscles working continuously for long periods (~10 to 30 min) [12]. Hence, in addition to the potent adaptive stimulus provided by lengthening contractions to the neuromuscular system [1], it has been suggested that ECC_CYC_ may also induce aerobic adaptations [16,22,55]. Our study provides novel information showing that the difference between ECC_CYC_ and CON_CYC_ training effects (i.e., net effect) on $\dot{V}O_{2\max}$ is small (µ = 2%), with CON_CYC_ being slightly more effective than ECC_CYC_. The meta-analysis of pre-to-post training effects showed 7% [95% CrI 0.5%, 12%] and 4% [95% CrI −7%, 15%] average increases in $\dot{V}O_{2\max}$ following CON_CYC_ and ECC_CYC_ training, respectively, which suggests that ECC_CYC_ can also positively affect aerobic fitness. Indeed, we found that $\dot{V}O_{2\max}$ enhancement is quite likely (80% of probability) following ECC_CYC_ training. Interestingly, group-level analysis of the population revealed greater probabilities of patients with cardiopulmonary disease (*p* < 0 = 69%) and obese adolescents (*p* < 0 = 58%) to benefit from greater increases in $\dot{V}O_{2\max}$ following ECC_CYC_ compared to CON_CYC_ training, while the estimated posterior densities for healthy individuals, amateur cyclists and sedentary participants showed smaller probabilities of greater increases in $\dot{V}O_{2\max}$ following ECC_CYC_ compared to CON_CYC_ training (*p* < 0 = 40%, 38%, and 10%, respectively). It is important to note that ECC_CYC_ and CON_CYC_ training protocols adopted in studies using sedentary participants were performed at the same PO [23,25], whereas the remaining sub-groups of group-level analysis of the population used ECC_CYC_ and CON_CYC_ training protocols performed at the same relative HR [19,24], at the same VO_2_ [20,43], or similar RPE [26,44,45]. Thus, as previously discussed in the literature [22], ECC_CYC_ sessions performed at the same PO as CON_CYC_ sessions may produce an insufficient stimulus to improve the aerobic capacity of some populations since ECC_CYC_ would be performed at a lower metabolic demand in such a condition.

The only meta-analyzed variable in this study for which CON_CYC_ training induced considerably greater increases compared to ECC_CYC_ was PPO. The results showed that the changes in PPO induced by CON_CYC_ were 7% [95% CrI 2%, 11%] greater than the changes induced by ECC_CYC_ training. Moreover, the posterior distribution of the estimated average effect indicated a 100% of probability of the CON_CYC_ being more effective than ECC_CYC_ in improving PPO. However, the meta-analysis of pre-to-post training effects showed an average increase of 11% [95% CrI 5%, 16%] following ECC_CYC_ training, with a posterior probability of 100% of ECC_CYC_ inducing a positive effect on PPO. Moreover, the meta-regression of conditional effects on PPO revealed an important effect of intervention duration in the difference between training modalities, which decreases as intervention duration increases. In other words, the effectiveness of the modalities in improving PPO tends to be similar as intervention duration increases. This may be associated with long-lasting muscle remodeling and delayed manifestation of gains in concentric muscle power following eccentric training regimens [56,57,58]. Nevertheless, the results obtained in the present study indicate that ECC_CYC_ can increase PPO achieved during a CON_CYC_ incremental test, but it is less effective than CON_CYC_ training in increasing this variable.

One of the most discussed applications of ECC_CYC_ is its implementation into exercise programs for special populations (i.e., frail individuals and patients with chronic diseases) [12,13]. In this context, the improvement in functional capacity is one of the main objectives of the exercise intervention. The distance covered in the six-minute walking test is an important measure of functional capacity and is associated with quality of life and longevity in older individuals and patients with chronic diseases [59]. The obtained data indicate that ECC_CYC_ training induces greater increases of 6MWD compared to CON_CYC_ training. Although the estimated difference between changes induced by ECC_CYC_ and CON_CYC_ on 6MWD was small (µ = 2%), the posterior distribution of the average population effect revealed a higher probability (*p* < 0 = 78%) of ECC_CYC_ training inducing a greater increase in 6MWD compared to CON_CYC_. The meta-analyses of pre-to-post training effects showed an average increase in 6MWD of 11% [95% CrI 7%, 15%] versus 8% [95% CrI 5%, 12%] following ECC_CYC_ and CON_CYC_ training, respectively, with both modalities presenting high probabilities of being effective in increasing 6MWD. However, most studies included in the meta-analysis of 6MWD effects reported lower cardiovascular burden [18,44,45] and lower sensation of dyspnea [14,45] during ECC_CYC_ compared to CON_CYC_ training sessions. This can be considered an important advantage of ECC_CYC_ interventions for exercise treatment of clinical populations. Importantly, all studies included in the meta-analysis of 6MWD effects involved patients with cardiopulmonary diseases. Therefore, the present evidence supports the utilization of ECC_CYC_ training for the development of functional capacity in patients with cardiopulmonary diseases.

The reduction in body fat content is an important outcome in the treatment of chronic diseases and obesity [60]. For this purpose, aerobic exercises such as running and CON_CYC_ are widely utilized [30]. The results of this study showed that ECC_CYC_ training may be as effective as CON_CYC_ training in reducing body fat percentage. The pre-to-post training average effects showed that both cycling modes induced ~1% reductions in BF%. The studies included in the meta-analysis of BF% effects were conducted with obese adolescents [20] and patients with cardiopulmonary diseases [34,35]. Thus, it is still unknown if ECC_CYC_ is effective in decreasing BF% in healthy, active individuals. It has been suggested that metabolic substrate utilization during eccentric exercise differs from concentric exercise, with increased fat oxidation rate and reduced glucose utilization during eccentric modalities [61]. Conversely, there is evidence showing similar utilization of energetic substrate between ECC_CYC_ and CON_CYC_ sessions when performed at the same VO_2_ [62]. The studies included in the analysis of BF% changes involved ECC_CYC_ and CON_CYC_ interventions performed with similar metabolic demands (i.e., VO_2_), which may be an important factor influencing the magnitude of BF% changes following ECC_CYC_ [22]. To date, it is unclear whether ECC_CYC_ is effective in decreasing BF% of all types of the population, but the present data support its utilization, at least for obese adolescents and patients with cardiopulmonary diseases.

## 5. Limitations

Some limitations should be considered when interpreting the current findings. The number of studies included in the meta-analysis of ICPT, IEPT, f-CSA, and BF% effects was small (4, 3, 3, and 3 studies, respectively), which may have affected the accuracy of the estimates of combined effects (i.e., average population effect) [63]. Moreover, the studies included in this review were conducted with different populations and training protocols. Thus, meta-regressions and group-level analyses were conducted to verify the impact of the population and intervention characteristics on effect sizes. However, meta-regression analyses were only possible for two variables (i.e., IPT and PPO), with $\dot{V}O_{2\max}$ and intervention duration as covariates, and group-level analyses were conducted considering the healthy condition of the participants and intervention duration as random effects. Hence, it was not possible to verify the impact of other populations and intervention characteristics in the chronic adaptations to ECC_CYC_ compared to CON_CYC_, such as the sex of the participants or the intensity used during training. Due to scarce evidence on the mechanisms underpinning the chronic adaptations to ECC_CYC_, it was necessary to interpret and discuss some of our results in light of evidence from other eccentric exercise modalities. Additionally, there is a lack of studies investigating the chronic effects of ECC_CYC_ in trained athletes and healthy, active individuals.

## 6. Future Perspectives

Currently, the positive benefits of eccentric exercise modalities in improving strength and muscle mass in different populations are widely recognized [2,3,10,12,51]. The present data extend our knowledge on the possible applications of eccentric muscle work, indicating that ECC_CYC_ training may be used as a time-effective modality to improve distinctive performance, physiological, and morphological parameters. The results of this study also indicate that the effectiveness of ECC_CYC_ training in inducing different adaptations, such as improving neuromuscular function and aerobic power at the same time, may be influenced by the fitness level of the participants and the intensity of the exercise sessions. Due to the unique force production-to-energy demand relationship of ECC_CYC_, the metabolic disturbances during low-demanding ECC_CYC_ sessions (e.g., ECC_CYC_ sessions performed at the same PO of CON_CYC_) may not be sufficient to trigger aerobic adaptations in sedentary individuals [25]. On the other hand, ECC_CYC_ prescribed at a metabolic load similar to that elicited by CON_CYC_ training was effective in improving both neuromuscular function and aerobic power parameters in clinical patients [19,43,45]. Future investigations should address the impact of different training intensities in the adaptations promoted by ECC_CYC_ training in distinct populations to optimize ECC_CYC_ prescription and individualization.

## 7. Conclusions

The current evidence indicates that ECC_CYC_ is a feasible modality for exercise interventions aiming to improve parameters of muscle strength, hypertrophy, functional capacity, aerobic power, as well as body composition, with greater effectiveness than CON_CYC_ training in improving strength-related variables. Additionally, ECC_CYC_ may be more advantageous than CON_CYC_ training in improving the aerobic power of patients with cardiopulmonary diseases. Therefore, ECC_CYC_ training can be integrated as a time-effective modality in exercise interventions aiming to improve key physical/physiological parameters associated with good quality of life and healthy aging, such as leg muscle strength, aerobic power, and whole-body fat content. Furthermore, ECC_CYC_ constitutes a feasible alternative to CON_CYC_ for exercise treatment of patients with cardiopulmonary disease given its tolerability and greater effectiveness.

## Figures and Tables

**Figure 1 ijerph-20-02861-f001:**
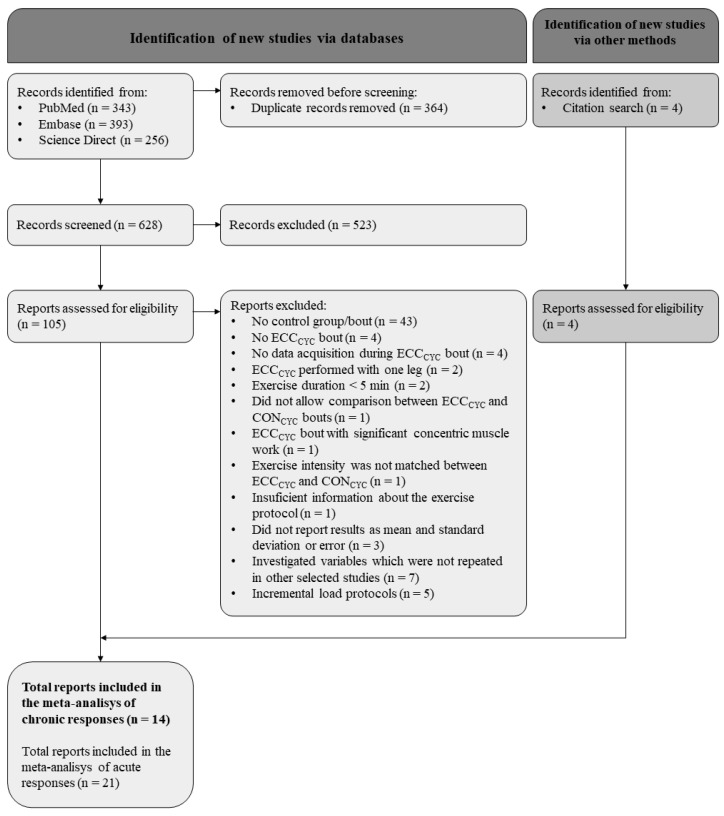
Flow diagram of the selection process of eligible studies. Abbreviations: CON_CYC_—concentric cycling; ECC_CYC_—eccentric cycling.

**Figure 2 ijerph-20-02861-f002:**
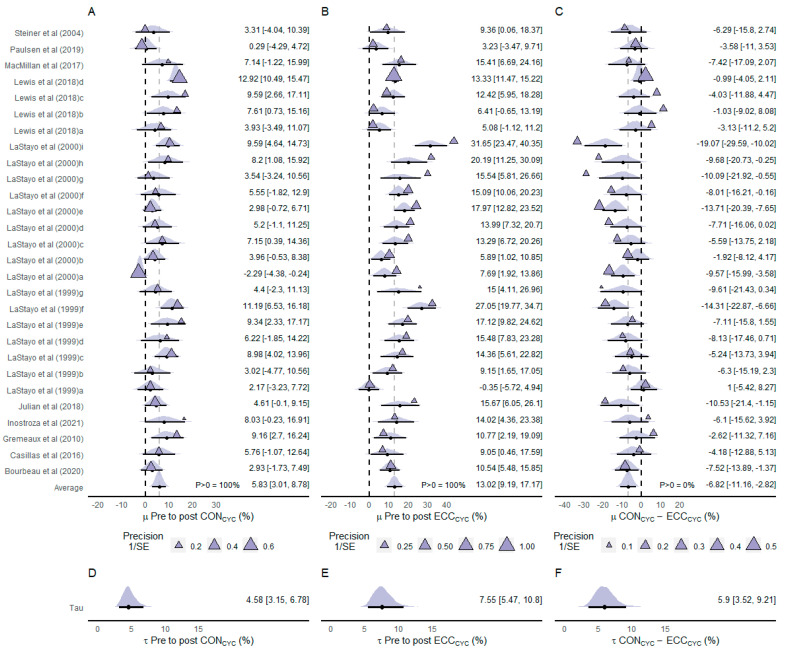
Forest plot of effect sizes (% mean difference) of CON_CYC_ training (**A**), ECC_CYC_ training (**B**), and net effect between training modalities (**C**) on isometric peak torque. Heterogeneity between effects of CON_CYC_ training (**D**), ECC_CYC_ training (**E**), and net effects (**F**). The densities represent model estimates (i.e., the posterior distribution). Black dots and whiskers are the posterior effect size median and 95% credible interval, respectively. The triangles are the studies’ observed mean effect sizes, and their sizes represent the precision of the effect, presented as the inverse of the standard error (1/SE), i.e., the larger the size of the triangle, the smaller the standard error. The letters a, b, c, and d in Lewis et al. (2018) [25] indicate assessments performed at different time points (after 3, 5, 7, and 8 weeks, respectively). The letters a, b, c, d, e, f, g, h, and i in LaStayo et al. (2000) [24] indicate assessments performed at different time points (after 1, 2, 3, 4, 5, 6, 7, 8, and 10 weeks, respectively). The letters a, b, c, d, e, f, and g in LaStayo et al. (1999) [46] indicate assessments performed at different time points (after 1, 2, 3, 4, 5, 6, and 7 weeks, respectively). Abbreviations: CON_CYC_—concentric cycling; ECC_CYC_—eccentric cycling. References: [14,19,20,24,25,26,34,35,44,45,46].

**Figure 3 ijerph-20-02861-f003:**
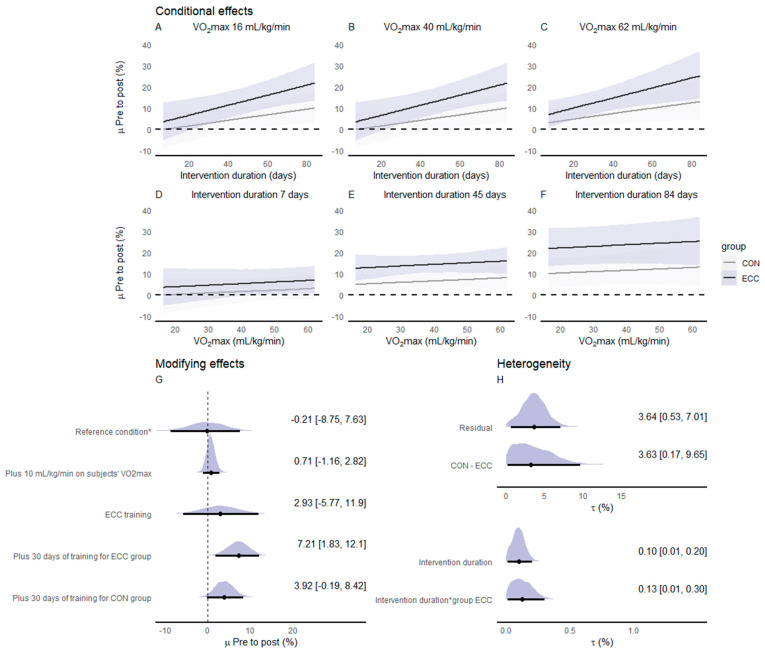
Meta-regression derived changes pre-to-post on CON_CYC_ and ECC_CYC_ conditions for isometric peak torque. Posterior medians with 95% credible intervals for conditional effects regarding the duration of the intervention adjusted for the minimum (**A**), mean (**B**), and maximum (**C**) values of $\dot{V}O_{2\max}$. observed on subjects of included studies and for $\dot{V}O_{2\max}$ adjusted to the minimum (**D**), mean (**E**), and maximum (**F**) values of intervention duration observed in the included studies. Modifying effects (posterior median [95% credible intervals]) in relation to the reference condition * (**G**). Heterogeneity as standard deviations (tau) values for random effects included in the meta-regression (**H**). Abbreviations: CON_CYC_—concentric cycling; ECC_CYC_—eccentric cycling; $\dot{V}O_{2\max}$—maximal oxygen uptake. * Reference condition adjusted to CON_CYC_, with 7 days of intervention, and subjects $\dot{V}O_{2\max}$ of 16 mL/kg/min.

**Figure 4 ijerph-20-02861-f004:**
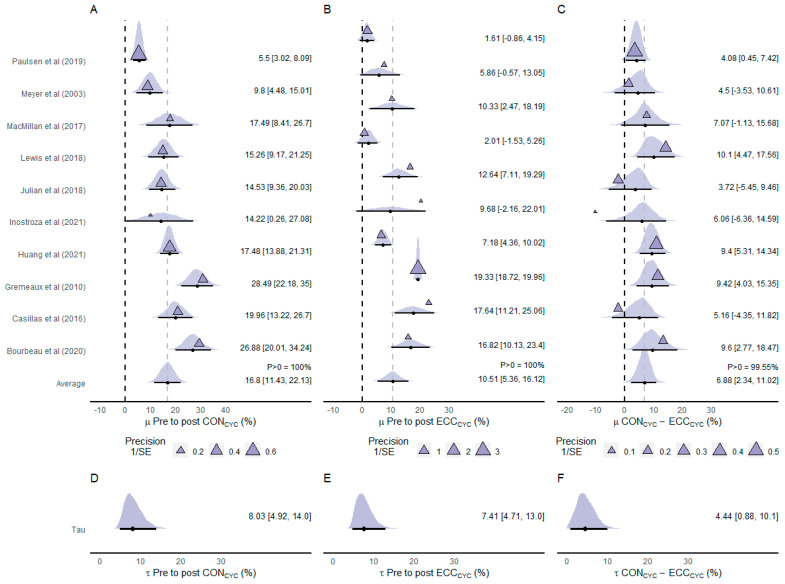
Forest plot of effect sizes (% mean difference) of CON_CYC_ training (**A**), ECC_CYC_ training (**B**), and net effect between training modalities (**C**) on peak power output. Heterogeneity between effects of CON_CYC_ training (**D**), ECC_CYC_ training (**E**), and net effects (**F**). The densities represent model estimates (i.e., the posterior distribution). Black dots and whiskers are the posterior effect size median and 95% credible interval, respectively. The triangles are the studies’ observed mean effect sizes, and their sizes represent the precision of the effect, presented as the inverse of the standard error (1/SE), i.e., the larger the size of the triangle, the smaller the standard error. Abbreviations: CON_CYC_—concentric cycling; ECC_CYC_—eccentric cycling. References: [14,19,20,23,25,26,35,43,44,45].

**Figure 5 ijerph-20-02861-f005:**
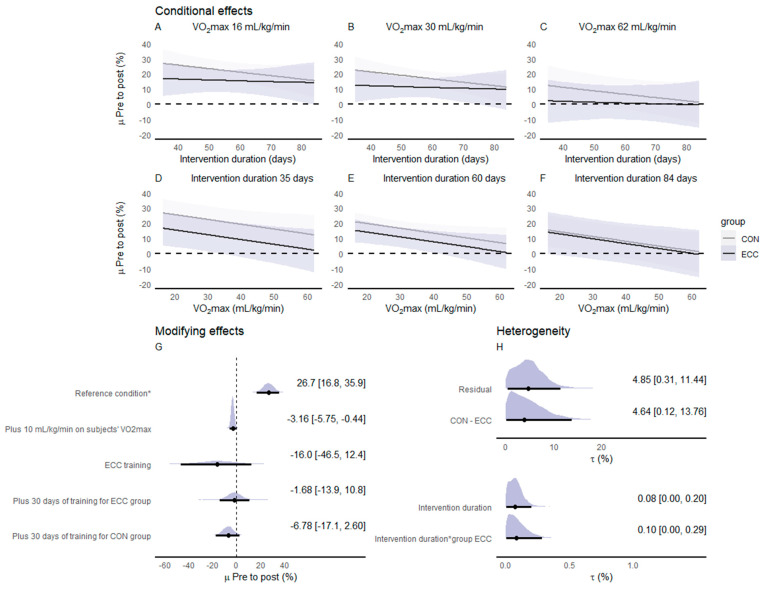
Meta-regression-derived changes pre-to-post on CON_CYC_ and ECC_CYC_ conditions for the peak power output of an incremental test. Posterior medians with 95% credible intervals for conditional effects regarding the duration of the intervention adjusted for the minimum (**A**), mean (**B**), and maximum (**C**) values of $\dot{V}O_{2\max}$. observed on subjects of the included studies and for $\dot{V}O_{2\max}$ adjusted to the minimum (**D**), mean (**E**), and maximum (**F**) values of intervention duration observed in the included studies. Modifying effects (posterior median [95% credible intervals]) in relation to the reference condition * (**G**). Heterogeneity as standard deviations (tau) values for random effects included in the meta-regression (**H**). Abbreviations: CON_CYC_—concentric cycling; ECC_CYC_—eccentric cycling; $\dot{V}O_{2\max}$—maximal oxygen uptake. * Reference condition adjusted to CON_CYC_, with 35 days of intervention, and subjects $\dot{V}O_{2\max}$ of 16 mL/kg/min.

**Figure 6 ijerph-20-02861-f006:**
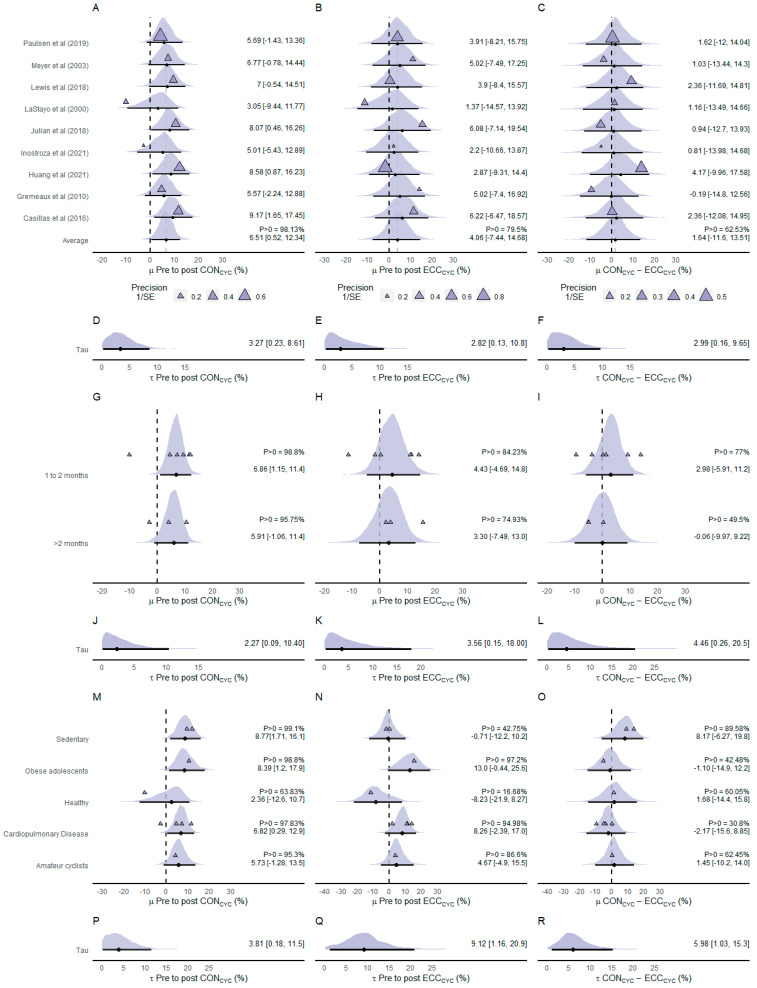
Forest plot of effect sizes (% mean difference) of CON_CYC_ training (**A**), ECC_CYC_ training (**B**), and net effect between training modalities (**C**) on maximal oxygen uptake. Heterogeneity between effects of CON_CYC_ training (**D**), ECC_CYC_ training (**E**), and net effects (**F**). Group-level effects of intervention duration of CON_CYC_ training (**G**), ECC_CYC_ training (**H**), and net effect between training modalities (**I**) on maximal oxygen uptake. Heterogeneity between group-level effects of intervention duration of CON_CYC_ training (**J**), ECC_CYC_ training (**K**), and net effects (**L**). Group-level effects of the population of CON_CYC_ training (**M**), ECC_CYC_ training (**N**), and net effect between training modalities (**O**) on maximal oxygen uptake. Heterogeneity between group-level effects of the population of CON_CYC_ training (**P**), ECC_CYC_ training (**Q**), and net effects (**R**). The densities represent model estimates (i.e., the posterior distribution). Black dots and whiskers are the posterior effect size median and 95% credible interval, respectively. The triangles are the studies’ observed mean effect sizes, and, in panels a, b, and c, their sizes represent the precision of the effect, presented as the inverse of the standard error (1/SE), i.e., the larger the size of the triangle, the smaller the standard error. Abbreviations: CON_CYC_—concentric cycling; ECC_CYC_—eccentric cycling. References: [19,20,23,24,25,26,43,44,45].

**Figure 7 ijerph-20-02861-f007:**
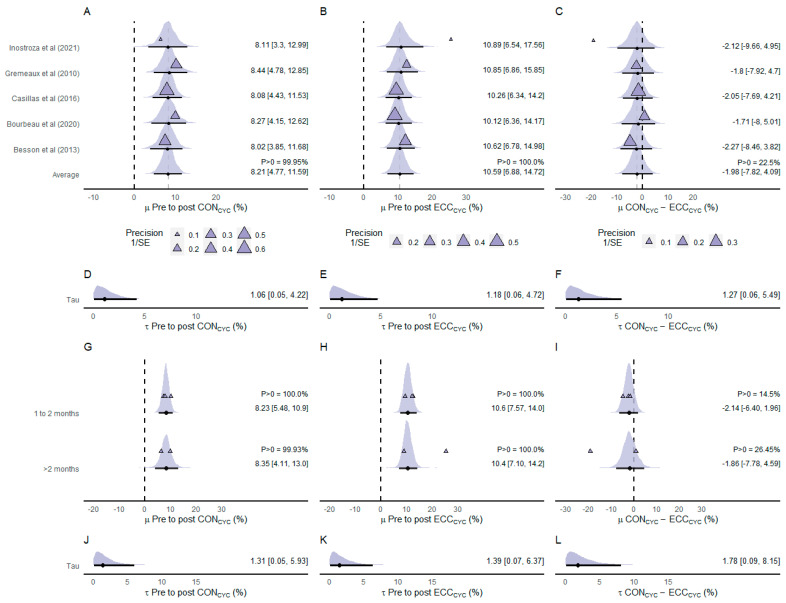
Forest plot of effect sizes (% mean difference) of CON_CYC_ training (**A**), ECC_CYC_ training (**B**), and net effect between training modalities (**C**) on six-minute walking distance. Heterogeneity between the effects of CON_CYC_ training (**D**), ECC_CYC_ training (**E**), and net effects (**F**). Group-level effects of intervention duration of CON_CYC_ training (**G**), ECC_CYC_ training (**H**), and net effect between training modalities (**I**) on six-minute walking distance. Heterogeneity between group-level effects of intervention duration of CON_CYC_ training (**J**), ECC_CYC_ training (**K**), and net effects (**L**). The densities represent model estimates (i.e., the posterior distribution). Black dots and whiskers are the posterior effect size median and 95% credible interval, respectively. The triangles are the studies’ observed mean effect sizes, and, in panels a, b, and c, their sizes represent the precision of the effect, presented as the inverse of the standard error (1/SE), i.e., the larger the size of the triangle, the smaller the standard error. Abbreviations: CON_CYC_—concentric cycling; ECC_CYC_—eccentric cycling. References: [14,18,19,44,45].

**Table 1 ijerph-20-02861-t001:** Study characteristics.

Study	Population	Intervention				Training Outcomes
		Training Period	Weekly Frequency	Session Duration	Intensity (ECC_CYC_/CON_CYC_)	
Besson et al. [18]	CHF	7 weeks	3 sessions	25 min	RPE between 9–11 on Borg scale/HR corresponding to VT	6MWT distance and the HR and VO_2_ during 6MWT
	patients					
	(n = 30)					
Bourbeau et al. [14]	COPD	10 weeks	3 sessions	30 min	Four times the PO corresponding to 60–80% PPO/60–80% PPO	Knee extensors IPT, ICPT, IEPT, and EPP, PPO, endurance time to CON_CYC_ at 75% PPO, and 6MWT distance
	patients					
	(n = 24)					
Casillas et al. [44]	CHF	7 weeks	3 sessions	25 min	RPE between 9–11 on Borg scale/HR corresponding to VT	Knee extensors and plantar flexors IPT, PPO, $\dot{V}O_{2\max}$, HR_max_, VT, and 6MWT distance
	patients					
	(n = 42)					
Gremeaux et al. [19]	CAD	5 weeks	3 sessions	30 min	HR corresponding to VT/HR corresponding to VT	Knee extensors and plantar flexors IPT, PPO, $\dot{V}O_{2\max}$, 6MWT distance, and 200-m fast walk test
	patients					
	(n = 14)					
Huang et al. [23]	Sedentary individuals	6 weeks	5 sessions	30 min	45–70% PPO/45–70% PPO	PPO, $\dot{V}O_{2\max}$, erythrocyte metabolic characteristics, and O_2_ release capacity
	(n = 24)					
Inostroza et al. [45]	COPD	12 weeks	2–3 sessions	2–3 × 10–15 min	RPE between 11–13 on Borg scale/RPE between 11–13 on Borg scale	Knee extensors IPT and RTD, body composition, $\dot{V}O_{2\max}$, 6MWT distance, timed up-and-go test, stairs ascending and descending walking time, and quality of life
	patients					
	(n = 20)					
Julian et al. [20]	Obese adolecents	12 weeks	3 sessions	30 min	20–70% $\dot{V}O_{2\max}$/20–70% $\dot{V}O_{2\max}$	Knee extensors IPT, ICPT, and IEPT, PPO, $\dot{V}O_{2\max}$, body composition, glycemia, insulinemia, and plasma levels of cholesterols and triglycerides
	(n = 23)					
LaStayo et al. [46]	Healthy individuals	6 weeks	2–5 sessions	10–30 min	100–300W during the first 3 wk, and then, PO was adjusted to match VO_2_ between modes/50–100W during the first 3 wk and then, PO was adjusted to match VO_2_ between modes	Knee extensors IPT
	(n = 9)					
LaStayo et al. [24]	Healthy individuals	8 weeks	2–5 sessions	15–30 min	54–65% HR_age_/54–65% HR_age_	Knee extensors IPT, $\dot{V}O_{2\max}$, HR_max_, f-CSA and ultrastructure characteristics, capilary-to-fiber ratio, and density
	(n = 13)					
Lewis et al. [25]	Sedentary individuals	8 weeks	2 sessions	20 min	60% PPO/60% PPO	Knee extensors IPT, leg press 6RM, PPO, $\dot{V}O_{2\max}$, HR_max_, and blood pressure
	(n = 17)					
MacMillan et al. [35]	COPD	10 weeks	3 sessions	30 min	Four times the PO corresponding to 60–80% PPO/60–80% PPO	Knee extensors IPT, PPO, body composition, f-CSA, mitochondrial function, and adaptation markers
	patients					
	(n = 15)					
Meyer et al. [43]	CAD	8 weeks	3 sessions	30 min	60% $\dot{V}O_{2\max}$ and/or 85% HR_max_/60% $\dot{V}O_{2\max}$ and/or 85% HR_max_	PPO, $\dot{V}O_{2\max}$, and central hemodynamic characteristics during ECC_CYC_ and CON_CYC_ sessions
	patients					
	(n = 13)					
Paulsen et al. [26]	Amateur	10 weeks	2 sessions	5–8 × 2 min	RPE between 12–17 on Borg scale/RPE between 12–17 on Borg scale	Knee extensors IPT, ICPT, and IEPT, muscle thickness, PPO, $\dot{V}O_{2\max}$, LT, 20-min CON_CYC_ time trial, CON_CYC_ Wingate test, and pedaling characteristics
	cyclists					
	(n = 23)					
Steiner et al. [34]	CAD	8 weeks	3 sessions	30 min	60% $\dot{V}O_{2\max}$/60% $\dot{V}O_{2\max}$	Knee extensors IPT, ICPT, and IEPT, body composition, and f-CSA
	patients					
	(n = 12)					

Abbreviations: 6MWT—6-min walking test; CAD—coronary artery disease; CON_CYC_—concentric cycling; COPD—chronic obstructive pulmonary disease; ECC_CYC_—eccentric cycling; EPP—eccentric peak power; f-CSA—vastus lateralis fiber cross-sectional area; HR—heart rate; HRage—age-predicted maximal heart rate; HR_max_—maximal heart rate; ICPT—isokinetic concentric peak torque; IEPT—isokinetic eccentric peak torque; IPT—isometric peak torque; LT—lactate threshold; PO—power output; PPO—peak power output; RPE—rate of perceived exertion; VO_2_—oxygen uptake; $\dot{V}O_{2\max}$—maximal oxygen uptake; VT—ventilatory threshold.

**Table 2 ijerph-20-02861-t002:** Physiotherapy Evidence Database (PEDro) score.

Study	PEDro Scale											Total Score
	1	2	3	4	5	6	7	8	9	10	11	
Besson et al. [18]	1	1	0	1	0	0	0	1	0	1	1	5
Bourbeau et al. [14]	1	1	1	1	0	0	1	1	1	1	1	8
Casillas et al. [44]	1	1	1	1	0	0	1	0	1	1	1	7
Gremeaux et al. [19]	1	1	0	1	0	0	1	1	1	1	1	7
Huang et al. [23]	1	1	0	1	0	0	0	1	1	1	1	6
Inostroza et al. [45]	1	0	0	1	0	0	0	1	1	1	1	5
Julian et al. [20]	1	1	0	1	0	0	1	1	1	1	1	7
LaStayo et al. [46]	0	1	0	1	0	0	0	1	1	1	1	6
LaStayo et al. [24]	0	1	0	1	0	0	0	1	1	1	1	6
Lewis et al. [25]	1	0	0	1	0	0	0	0	0	1	1	3
MacMillan et al. [35]	1	1	0	1	0	0	1	1	0	1	1	6
Meyer et al. [43]	1	1	0	1	0	0	0	1	1	1	1	6
Paulsen et al. [26]	0	1	0	1	0	0	0	1	1	1	1	6
Steiner et al. [34]	0	1	0	1	0	0	0	1	1	1	1	6

## Data Availability

The data that support the findings of this study are available from the corresponding author upon reasonable request.

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
