# Peer review of "Chronic Adaptations to Eccentric Cycling Training: A Systematic Review and Meta-Analysis"

_ijerph, 2023, doi:10.3390/ijerph20042861_

Round 1

Reviewer 1 Report

Dear authors,

I reviewed your article  - Chronic Adaptations to Eccentric Cycling Training: A Systematic Review and Meta-Analysis - which aim was to investigate the effects of eccentric cycling (ECCCYC) training on performance, physiological, and morphological parameters in comparison to concentric cycling (CONCYC) training.

I appreciate the quality of your work and the deep analysis that you made in your meta-review concerning eccentric contraction and its effects: IPT (concentric, eccentric, isometric), PPO, maximal oxygen uptake, body fat.

Also, there are some limitations in the analysis of some items, concerning the number of study included, but there are in the limit of agreement.

I agree to endorse the publication of your article.

Author Response

R: We would like to thank you for taking time and effort to review our manuscript. We agree that the number of studies in the meta-analysis of isokinetic concentric and eccentric torque, fiber cross-sectional area, and body fat percentage is low and this limitation was highlighted in the section “Limitations” (lines 767-770).

Reviewer 2 Report

The manuscript was written smoothly, the literature was well cited, and had reference value.

Only this reference was required to confirm the year of citation.

LaStayo, P. C.; Reich, T. E.; Urquhart, M.; Hoppeler, H.; Lindstedt, S. L. Chronic eccentric exercise: improvements in muscle strength can occur with little demand for oxygen. Am J Physiol 2000, 276, R611-R615.

Author Response

R: We would like to thank you for taking time and effort to review our manuscript. We just revised this citation in the reference list. Please find the changes in lines 927-928.

Reviewer 3 Report

Congratulations, it is a work of great actuality and with a great scientific rigor, only thought that it can be improved I insinuate:

It is important to reduce the length of the article without losing its essence, I propose to reduce between 10 and 20% of the length, writing more succinctly and presenting the graphics in the form of a compendium. (One possibility to reduce the length of the text could be: From page 10 to 29 is the Meta-analyses, it could be to show only the highly significant and the remaining graphs and data leave them as appendices, available in some link or mail from the authors.)

It should be included in the limitations the fact of having taken concepts of general eccentric work to assume what can occur with eccentric cycling or eliminate these considerations. 

Author Response

R: We would like to thank you for taking time and effort to review our manuscript. We choose to present the forest plots of ICT, IET, f-CSA, and BF% as Supplementary Material since the number of included studies investigating these variables was small. This change has reduced by 15% the length of the manuscript. We also added information in the Limitations section regarding the concepts of general eccentric work used in the interpretation and discussion of our results. Please find this information added to lines 779-782.

Reviewer 4 Report

Dear authors,

the manuscript presents a good systematic review of chronic adaptations to eccentric cycling training.

However, it looks confusing in some passages. Too many graphics and finally, the text is too long. I, therefore, recommend streamlining the text and/or simplifying the graphics and captions.

Author Response

R: We would like to thank you for taking time and effort to review our manuscript. We choose to present the forest plots of ICT, IET, f-CSA, and BF% as Supplementary Material since the number of included studies investigating these variables was small. This change has reduced by 15% the length of the manuscript.